# UHPLC-Q-Orbitrap-MS-Based Metabolomics Reveals Chemical Variations of Two Types of Rhizomes of *Polygonatum sibiricum*

**DOI:** 10.3390/molecules27154685

**Published:** 2022-07-22

**Authors:** Gelian Luo, Junhan Lin, Weiqing Cheng, Zhibin Liu, Tingting Yu, Bin Yang

**Affiliations:** 1Fujian Vocational College of Bioengineering, Fuzhou 350007, China; ggluo562@163.com (G.L.); cwqdog@126.com (W.C.); yutt3245@163.com (T.Y.); 2Institute of Food Science & Technology, Fuzhou University, Fuzhou 350108, China; liuzhibin@fzu.edu.cn; 3Nanwuyi Professional Cooperative of Chinese Herbal Medicine Planting, Shaowu 354000, China; yangbin7211@126.com

**Keywords:** rhizomes of *Polygonatum sibiricum*, evergreen, metabolomics, chemical markers, UHPLC-Q-Orbitrap-MS

## Abstract

The rhizomes of *Polygonatum sibiricum* are commonly consumed as food and also used as medicine. However, the metabolic profile of *P. sibiricum* has not been fully revealed yet. Recently, we developed a novel evergreen species of P. sibiricum. The objectives of this study were to compare the metabolic profiles of two types of *P. sibiricum*, i.e., the newly developed evergreen type (Gtype) and a wide-type (Wtype), by using UHPLC-Q-Orbitrap-MS-based untargeted metabolomics approach. A total of 263 and 258 compounds in the positive and negative modes of the mass spectra were tentatively identified. Distinctively different metabolomic profiles of these two types of *P. sibiricum* were also revealed by principal component analysis (PCA) and principal coordinates analysis (PCoA). Furthermore, by using partial least squares discriminant analysis (PLS-DA) modeling, it was found that, as compared with Wtype, Gtype samples had significantly higher content of oxyberberine, proliferin, alpinetin, and grandisin. On the other hand, 15 compounds, including herniarin, kaempferol 7-neohesperidoside, benzyl beta-primeveroside, vanillic acid, biochanin A, neoschaftoside, benzyl gentiobioside, cornuside, hydroxytyrosol-glucuronide, apigenin-pentosyl-glucoside, obacunone, 13-alpha-(21)-epoxyeurycomanone, vulgarin, digitonin, and 3-formylindole, were discovered to have higher abundance in Wtype samples. These distinguishing metabolites suggest the different beneficial health potentials and flavor attributes of the two types of *P. sibiricum* rhizomes.

## 1. Introduction

The plant *Polygonatum sibiricum,* belonging to the family *Liliaceae*, is distributed widely in the northern temperate zone. Its leaves, stems, and rhizomes are commonly used as food in China. For example, the rhizomes can be fried with sugar and honey to make sweet snacks [1]. The rhizomes are also used to make tea or soaked in wine or liquor to flavor the beverages [1,2]. Moreover, the dried rhizomes of *P. sibiricum* have long been used as traditional herbal medicine in China, Korea, and Japan, for the treatment of various diseases, such as cough, fatigue, poor appetite, dizziness, and lung trouble [1,3]. Pharmacological studies also demonstrated the antioxidant [4], antimicrobial [5], anti-inflammatory [6], antiatherosclerotic [7], antidiabetic [8], antihyperglycemic [9], and anticancer [10] activities of *P. sibiricum.* Due to its popularity as food and potency as medicine, *P. sibiricum* is approved as “medicine food homology” by China’s Ministry of Health, which means that it combines the function of food and medicine, and can be used both for food and medicine [11].

The pharmacological activities of *P. sibiricum* have been attributed to various phytochemicals existing in the plant, such as phenolics [12], steroid saponins [13,14], and alkaloids [15]. For example, a phenolic glycoside from *P. sibiricum* identified by Chen et al. [16] showed potent inhibitory activity against α-glucosidase. The saponin extracts of *P. sibiricum* alleviated the symptoms of diabetes and regulated the gut microbiota of diabetic mice [9]. Additionally, polysaccharides isolated from *P. sibiricum* have also been reported to show various bioactivities [17]. Currently, chemical composition studies of *P. sibiricum* are mainly focused on selected categories of chemicals. However, like other natural functional foods and traditional herbal medicines, the constituents of *P. sibiricum* are complex, and yet undisclosed fully. Investigations into the profiling of chemicals of *P. sibiricum,* and screening of chemical markers of *P. sibiricum* derived from different sources remain inadequate.

Untargeted metabolomic approaches, especially those based on the ultra-high-performance liquid chromatography and Orbitrap high-resolution mass spectrometry (UHPLC-Q-Orbitrap-MS) combing with chemometric tools, e.g., principal component analysis (PCA), hierarchical cluster analysis (HCA), and partial least squares discriminant analysis (PLS-DA), have the advantage of unbiased detection and extensive coverage. UHPLC-Q-Orbitrap-MS-based metabolomics have been applied in the analysis of metabolites of various plant materials, such as *Camilla sinensis* [18], *Panax quinquefolius* [19], and *Olea europaea* [20]. Comprehensive chemical profiling of *P. sibiricum,* and additionally, elucidation of the biomarkers for distinguishing different genotypes and phenotypes of *P. sibiricum* could be achieved by the UHPLC-Q-Orbitrap-MS-based metabolomics approach.

Recently, we developed a novel evergreen hardy species of *P. sibiricum,* which grows and remains green in winter. Systematic and comprehensive elucidation of the phytochemical constituents of the novel species of *P. sibiricum* is of crucial importance in exploring the bioactivity basis and elaborating the quality standards to promote its modernization. In this study, we analyzed and compared the metabolic profiles of the rhizomes from two types of *P. sibiricum*, i.e., the newly developed evergreen type and wide-type, by UHPLC-Q-Orbitrap-MS combing with several chemometric tools. We also attempted to characterize the biomarkers for distinguishing these two types of *P. sibiricum.* The results of this study will improve the in-depth understanding of the chemical profiles and pharmacological actions of *P. sibiricum,* and thus may lay a foundation for quality control and the future use of this plant material in food and medicine.

## 2. Results and Discussion

### 2.1. Method Validation

In order to validate the repeatability and stability of the analytical method used in this study, three injections of the QC sample were implemented. The base peak chromatograms (BPCs) of the QC sample in both positive and negative ionization modes are demonstrated in Appendix A. The BPCs of the three injections were well overlapped, suggesting no obvious analytical variation in terms of retention time shift or intensity deviation. Furthermore, the Pearson correlation coefficients between the three injections of the QC sample were calculated based on the intensities of detected peaks. As shown in Appendix A, high correlations (all R^2^ values were higher than 0.95) were obtained, indicating satisfying stability of the detection process and data quality. Additionally, the relative standard deviation (RSD, %) of the intensities of the internal standard of 2-chlorophenylalanine in QC samples were calculated as 8.09% and 3.42% for positive and negative ionization modes, respectively, suggesting good signal repeatability and stability. Taken together, the results indicated that the methodology we employed was reliable.

### 2.2. Metabolite Identification

To comprehensively evaluate the chemical constituents of the dried rhizomes of *P. sibiricum*, a 75% aqueous methanol solution assisted with sonication was used to extract both polar and nonpolar compounds. Spectra of the 10 *P. sibiricum* samples from the two different types were acquired using UHPLC-Q-Orbitrap-MS under positive and negative ionization modes. The representative total ion chromatograms (TICs) of the two types of *P. sibiricum* are shown in Figure 1. Abundant information on metabolites can be observed in both ionization modes. Compared with the negative ionization mode, the positive mode illustrated more observable peaks. After chromatographic peak detection, sample alignment, and peak correspondence by using the XCMS package, a total of 52,575 and 46,211 features were obtained in the 10 *P. sibiricum* samples under positive and negative ionization modes, respectively. Based on the in-house and public metabolites databases, 263 and 258 compounds in the positive and negative modes of the mass spectra were tentatively identified. Further multivariate statistical analyses were mainly based on these tentatively identified metabolites and their corresponding semi-quantitative information, i.e., the integrated peak areas.

These compounds can be classified into 21 categories, including alkaloids, amino acid derivatives, benzene and substituted derivatives, carbohydrates and derivatives, chalcones, coumarins and derivatives, fatty acids, fatty acyls, flavonoids, iridoids, lignans, lipids, organic acids and derivatives, organic nitrogen compounds, organic oxygen compounds, phenolic acids, phenols, phenylpropanoids, quinones, terpenoids, and miscellaneous. The details on their retention time, tentative identification, chemical class, molecular formulas, accurate masses [M+H]^+^ or [M−H]^−^, and average peak areas in the two types of samples are shown in Appendix A. Heatmaps of these metabolites based on their integrated peak areas are demonstrated in Figure 2.

Alkaloids, terpenoids, and phenolic compounds that included flavonoids, phenolic acids, phenols, and phenylpropanoids, were the most abundant categories across all samples. Phenolic compounds are one of the most important groups of secondary metabolites in plants and are also well known for their multiple pharmacological activities. Phenols can be further classified into flavanols, flavonols, isoflavonols, flavanones, hydroxybenzoic acids, hydroxycinnamic acids, etc. [21]. Several flavonoid glycosides have been isolated from the genus of *Polygonatum*. For example, six flavonoid glycosides, including narcissoside, nicotiflorin, rutin, isovitexin, desmodin, and saponarin, have been reported to exist in *Polygonatum odoratum* [22]. In this study, isovitexin, nicotiflorin, and runtin were also identified in the rhizomes of both types of *P. sibiricum* samples using UHPLC-Q-Orbitrap-MS under the negative ionization mode. Jin et al. [23] have reported the existence of four steroidal saponins, namely, polygonatosides A−D, in the rhizomes of *Polygonatum zanlanscianense*. Additionally, ten furostanol saponins have been isolated from the rhizomes of *Polygonatum kingianum*, including kingianoside D, kingianoside C, etc. [24]. In this study, we also identified polygonatumoside D and kingianoside C in the rhizomes of *P. sibiricum*. Regarding the alkaloids, Sun et al. [15] isolated and identified two alkaloids, namely polygonatine A and polygonatine B, from the rhizomes of *P. sibiricum*. In this study, the alkaloid polygonatine A was also identified, which is consistent with the previous reports. In general, an extensive amount of research work has been done on the elucidation of the chemical composition of this genus to date, the isolated compounds mainly belong to flavonoids, terpenoids, alkaloids, as well as other categories such as amino acids and lipids. However, most of these studies focus on relatively limited number of chemical compounds or on some specific chemical categories. This study provides a more comprehensive insight into the metabolic profile of the rhizomes of *P. sibiricum.*

### 2.3. Metabolite Profiles Comparison and Differentiating Metabolites Analysis

Due to the chemical complexity, the differences between the two types of *P. sibiricum* were not clear from UHPLC-Q-Orbitrap-MS chromatograms. Therefore, their intrinsic similarities and differences were inspected by using PCA and PCoA. In both positive and negative ionization modes, the scatter plots of PCA showed that all samples were clearly separated into two different groups, which was consistent with the types of samples (Figure 3A,B). Similarly, the scatter patterns in PCoA also confirmed the results obtained from PCA (Figure 3C,D). Additionally, the hierarchical clustering dendrograms of all samples, as shown in Figure 2 (above the heatmaps), also exhibited two major clusters, consistent with their types. Therefore, these results revealed the distinctive differences in metabolite profile between the Wtype and Gtype of *P. sibiricum*.

Next, volcano plots were used to visualize the differentiating metabolites between the two types of samples (Figure 4). Metabolites with fold change ≥2.0 or ≤0.5 and *p*-value < 0.01 were regarded as differentially altered metabolites, and thus colored as red (significantly higher in Gtype samples, fold change ≥ 2.0 and *p*-value < 0.01) or green (significantly higher in Wtype samples, fold change ≤ 0.5 and *p*-value < 0.01). It can be seen that in the positive ionization mode, 27 metabolites were identified as the differentially altered metabolites, of which 10 metabolites were significantly higher in Gtype samples and 17 metabolites were significantly higher in Wtype samples (Figure 4A). With regard to the negative ionization mode, 24 metabolites were identified as the differentially altered metabolites, of which 8 metabolites were significantly higher in Gtype samples and 16 metabolites were significantly higher in Wtype samples (Figure 4B).

In order to further identify the biomarkers that better distinguish the two types of *P. sibiricum* samples, the supervised PLS-DA modeling based on the 27 differentially altered metabolites that were identified in positive ionization mode or 24 differentially altered metabolites that were identified in negative ionization mode was utilized. As shown in the PLS-DA score plots, the samples were clearly distinct and in line with their types (Figure 5A,C). In cross-validation of the PLS-DA model for positive ionization mode, the model fit (R^2^) and predictiveness (Q^2^) values were both found to be 0.99, suggesting good fitness and predictive power of this model. For negative ionization mode, R^2^ and Q^2^ values were 0.98 and 0.99, respectively. Next, VIP scores were calculated to further assess the contribution of each metabolite to the total variance in the constructed PLS-DA models. The results are shown in Figure 5B,D. A range of metabolites (15 metabolites for positive ionization mode and 11 metabolites for negative ionization mode) was selected as the biomarkers, on the basis that the VIP values were higher than 1.0. Of these biomarkers, l-tryptophan was the overlapping metabolite in both positive and negative ionization modes. Therefore, a total of 25 metabolites were thus identified as the biomarkers for distinguishing the two types of *P. sibiricum* samples. Among them, 12 metabolites belonged to phenolic compounds, including alpinetin, grandisin, herniarin, kaempferol 7-neohesperidoside, benzyl beta-primeveroside, vanillic acid, biochanin A, neoschaftoside, benzyl gentiobioside, cornuside, hydroxytyrosol-glucuronide, and apigenin-pentosyl-glucoside. Five metabolites were tentatively identified as terpenoids, including proliferin, obacunone, 13-alpha-(21)-epoxyeurycomanone, vulgarin, and digitonin. Additionally, 2 metabolites were tentatively identified as alkaloids, including oxyberberine and 3-formylindole. In addition, two amino acids (cystine and l-tryptophan), two DNA nucleotide bases (thymidine and adenine), one glycerol derivative (gingerglycolipid A), and one small molecule sugar (sucrose/maltose) were identified. Due to the potential bioactivities of phenolic compounds, terpenoids, and alkaloids, the metabolites belonging to these three classes were further inspected.

### 2.4. Comparison of the Differentiating Phenolic Compounds, Terpenoids, and Alkaloids

The comparison of the peak areas of the differentiating twelve phenolic compounds, five terpenoids, and two alkaloids is demonstrated in Figure 6. In specific, four metabolites, including oxyberberine, proliferin, alpinetin, and grandisin, showed higher abundance in Gtype samples (Figure 6A). As shown in Figure 6A, approximately 24 times higher amounts of oxyberberine were detected in Gtype *P. sibiricum* samples under the positive ionization mode than those in Wtype *P. sibiricum* samples. The representative extracted ion chromatograms of oxyberberine (m/z 352.11555) and its fragmentation spectrum are shown in Figure 7A,B. Oxyberberine is a natural alkaloid found in many plants, such as *Thalictrum alpinum* L. [25], *Coptidis chinensis* Franch [26], *Berberis lycium* Royle [27], and *Phellodendron chinense* Schneid [28]. Oxyberberine is an oxoderivative of berberine. Studies have demonstrated that this compound has a number of pharmacological activities, including the anti-proliferative effects against a series of cancers [27], and anti-inflammation effects by reducing the levels of TNF-α, IL-1β, IL-6, etc. [26], hypoglycemic effect via regulating the PI3K/Akt and Nrf2 signaling pathways [29], and an antiarrhythmic activity that is analogous to class III antiarrhythmic agents [30]. Moreover, as reported by Li et al., oxyberberine exhibited more pronounced anti-inflammatory activities than its precursor berberine [26]. Due to a broad array of pharmacological activities, oxyberberine has promising pharmacological potential. In this study, we also identified another berberine derivative, epiberberine. Interestingly, the Gtype samples also had a higher abundance of epiberberine as compared with the Wtype samples. In specific, the average peak area of this compound in Gtype samples was approximately 5.6 times higher than that in Wtype samples. The representative extracted ion chromatograms of epiberberine (m/z 336.12009) and its fragmentation spectrum are demonstrated in Figure 7C,D. Epiberberine is also a natural protoberberine alkaloid found in several plants like *Sinomenium acutum* [31], *Coptis chinensis* Franch [32], and *Corydalis turtschaninovii* [33]. Epiberberine is a multi-target small molecule with a low toxicity and exerts many activities [32]. For example, it has been reported to exert antioxidant and anti-Alzheimer activities [34]. Liu et al. [32] has systematically reviewed the therapeutic effects of epiberberine, and concluded that epiberberine exerted beneficial effects in various diseases. Therefore, the higher content of oxyberberine and epiberberine in the newly developed evergreen type of *P.*
*sibiricum* may make it have potent application potential in functional food and medicine. The underlying mechanism of higher abundance of oxyberberine and epiberberine in the evergreen type of *P.*
*sibiricum* is currently still unknown, further studies are warranted to elucidate such changes.

Two phenolic compounds, alpinetin and grandisin, were also identified as the biomarkers whose abundances were significantly higher in Gtype samples. As revealed by UHPLC-Q-Orbitrap-MS, approximately 12 times higher alpinetin and 4.2 times higher grandisin were detected in Gtype samples as compared with those in Wtype samples. The representative extracted ion chromatograms of alpinetin (m/z 293.07944) and grandisin (m/z 415.21112), as well as their respective fragmentation spectra, are demonstrated in Figure 8. Alpinetin (7-hydroxy-5-methoxyflavanone; C_16_H_14_O_4_), a natural dihydroflavone, was found in many medicinal herbs, such as *Alpinia intermedia* Gagnep [35], *Amomum subulatum* Roxb. [36], *Carya cathayensis* Sarg [37]. This natural flavonoid has also been reported to exist in *Polygonatum ferruginea* [38]. Alpinetin is one of the major active compounds of Chinese patent drugs such as Jianweizhitong tablet, Fufangcaodoukou tincture, Baikoutiaozhong pill, and Xingqiwenzhong granule [39]. Recently, Zhao et al. [39] systematically reviewed the pharmacological activities of alpinetin, including the counteraction against cardiovascular disease, inflammatory diseases, carcinoma, bacterial and virus infection, and neuro-disorders. Regarding grandisin, it is a tetrahydrofuran neolignan, isolated from *Piper solmsianum* [40], and *Virola surinamensis* [41], with the pharmacological activities of antinociception and anti-inflammation [41]. Hence a broad spectrum of biological activities of alpinetin and grandisin has been recognized. Herein, we demonstrated the higher abundance of these two bioactive phytochemicals existing in the evergreen type of *P.*
*sibiricum*, suggesting important therapeutic applications of this new phenotype of *P. sibiricum* species.

In addition to oxyberberine, alpinetin, and grandisin, proliferin, a bicyclic sestertepene, was also identified as the biomarker in distinguishing the two types of *P. sibiricum*. However, information regarding its existence in plants and its pharmacological activity is currently lacking. Therefore, this compound was classified as the “potential biomarker” for distinguishing the two types of *P. sibiricum*.

On the other hand, 15 metabolites, including obacunone, herniarin, 13-alpha-(21)-epoxyeurycomanone, kaempferol 7-neohesperidoside, 3-formylindole, vulgarin, benzyl beta-primeveroside, vanillic acid, biochanin A, neoschaftoside, benzyl gentiobioside, cornuside, digitonin, hydroxytyrosol-glucuronide, and apigenin-pentosyl-glucoside showed higher abundance in Wtype samples (Figure 6B). Many of them are also considered medicinal compounds with various pharmacological activities. For example, obacunone, a highly oxygenated terpenoid belonging to the class of limonoids, has been demonstrated for various biological activities including anticancer and anti-inflammation [42]. Cornuside, a secoiridoid glucoside found from the fruit of *Cornus officinalis* Sieb. et Zucc., has been proven to have anti-inflammatory effects [43]. Additionally, some of these compounds show a strong odor. For instance, vanillic acid, a phenolic acid found in vanilla and many other plants, has a milky, sweet, creamy, and pleasant, odor. Herniarin, belonging to the class of coumarins and derivatives, has a sweet, balsamic, and tonka-like taste. Therefore, the alteration in the concentration of these compounds in the evergreen type of samples may result in the changes in bioactivities and general flavor profile of the final product, when compared with the wide-type species.

## 3. Materials and Methods

### 3.1. Chemicals

Methanol, acetonitrile, and formic acid of LC-MS grade were purchased from CNW Technologies (Düsseldorf, Germany). 2-Chlorophenylalanine was purchased from HC Biotech (Shanghai, China). Ultrapure water (18.2 MΩ cm at 25 °C) was prepared using a Milli-Q water purification system (Millipore, Billerica, MA, USA).

### 3.2. Sample Preparation and Extraction

Two types of *P. sibiricum,* including evergreen type (Gtype) and wide-type (Wtype), were collected for analysis in this study. Both LRT and WT were cultured and grown by Shaowu City (Fujian, China). The basic information regarding the two types of plants is summarized in Appendix A. The fresh rhizomes of both types of *P. sibiricum* were collected and roughly cut into small pieces. After freeze-drying, samples were crushed into powder. An accurately weighed 100 mg sample was added to 500 μL of extracting solution, which was the 4:1 water–methanol mixture containing 1 μg/mL of the internal standard of 2-chlorophenylalanine. After 30 s vortex, the mixture was sonicated for 1 h in an ice-water bath for the extraction of the metabolites. After stilly placing for 1 h at −40 °C, the mixture was centrifuged (13,800× *g*, 15 min, 4 °C). A total of 300 μL of supernatant was collected and put in a fresh 2 mL Eppendorf tube for LC-MS analysis. The quality control (QC) sample was prepared by mixing an equal aliquot of the supernatants from all samples.

### 3.3. UHPLC-Q-Orbitrap-MS Analysis

Separation of the *P. sibiricum* extracts was performed on an Agilent ultra-high-performance liquid 1290 UPLC system (Agilent, Santa Clara, USA) equipped with a binary pump, split loop autosampler, column compartment, and diode array detector (range 190–680 nm). An Acquity UHPLC Ethylene Bridged Hybrid (BEH) C18 column (150 mm × 2.1 mm, 1.7 μm; Waters, Milford, MA, USA) with an Acquity UHPLC BEH C18 VanGuard guard column (5 mm × 2.1 mm, 1.7 μm; Waters, Milford, MA, USA) was used to perform the chromatographic separation. The sample injection volume was 5.0 μL. Mobile phases consisting of 0.1% (*v/v*) formic acid in water (A) and 0.1% (*v/v*) formic acid in acetonitrile (B) were used to elute the column, at a flow rate of 400 μL/min. The multi-step linear elution gradient program was set as follows: 0–3.5 min, 95–85% A; 3.5–6 min, 85–70% A; 6–6.5 min, 70–70% A; 6.5–12 min, 70–30% A; 12–12.5 min, 30–30% A; 12.5–18 min, 30–0% A; 18–25 min, 0–0% A; 25–26 min, 0–95% A; 26–30 min, 95–95% A.

After chromatographic separation, a Q Exactive Focus Orbitrap mass spectrometer (Thermo Fisher Scientific, Bremen, Germany) coupled with Xcalibur software was employed to obtain the MS and MS/MS data based on the information-dependent acquisition (IDA) mode. The mass spectrometer was operated in both positive and negative modes. The electrospray ionization (ESI) parameters were set as follows: spray voltage, 4.0 kV (positive) or −3.6 kV (negative); sheath gas (nitrogen) flow rate, 45 arbitrary units; auxiliary gas (nitrogen) flow rate, 15 arbitrary units; capillary temperature, 400 °C; resolution, MS full scan 70,000 full width at half maximum (FWHM), MS/MS scan 17,500 FWHM; scan range, m/z 100 to 1500. An external calibration for mass accuracy was carried out before the analysis according to the manufacturer’s guidelines.

### 3.4. Data Processing

Raw data of all samples acquired from the UHPLC-Q-Orbitrap-MS was first converted to the mzXML format by msConvert software and then processed by using the XCMS package in R software for peak extraction, peak alignment, and peak integration. Data processing was performed in both positive and negative ionization modes in order to analyze them individually. Based on the in-house metabolite database (Shanghai Biotree Biotech Co., Ltd., Shanghai, China) and the public database, the detected ion features were qualitatively analyzed. The semi-quantitative information of all tentatively identified compounds was used for further chemometric analysis, including principal component analysis (PCA), principal coordinates analysis (PCoA), and partial least squares discriminant analysis (PLS-DA) by using the ade4 package in R software. PCA and PCoA, the unsupervised pattern recognition methods, were employed to reduce the dimension of the data and reveal the intrinsic correlation of samples. PLS-DA, a supervised pattern recognition method, was utilized to maximize the separation of the metabolomic profiles between the two types of *P. sibiricum*. Furthermore, the variable importance in the projection (VIP) values were calculated to identify the metabolites that most significantly contributed to the discrimination of the metabolomic profiles between the two types of *P. sibiricum* in the PLS-DA model. The variables with VIP values ≥1.0 and *p*-value (student’s test) <0.01 were selected and utilized for further data analysis.

## 4. Conclusions

In this study, an untargeted metabolomic approach based on UHPLC-Q-Orbitrap-MS was employed to reveal the chemical variation of Gtype and Wtype *P. sibiricum* rhizomes. A total of 263 and 258 compounds in the positive and negative modes of the mass spectra were tentatively identified. Of these metabolites, phenolic compounds, terpenoids, and alkaloids were the most abundant categories across all samples. Furthermore, as revealed by PCA and PCoA, distinctive differences in metabolic profiles between these two types of samples were found. Additionally, we identified 19 differentiating phenolic compounds, terpenoids, and alkaloids by PLS-DA modeling. Because these two types of *P. sibiricum* rhizomes are varieties with the same genetic background, it is speculated that some of the differentiating metabolites are caused by differences in gene and protein expression. Therefore, further transcriptome and proteome studies to explore the differentiating gene and protein are warranted. These findings provide a reference for the subsequent analysis of the complex underlying mechanism of the variations in metabolic profile and provide useful information for the future application of the newly developed evergreen type *P. sibiricum* in functional food and medicine.

## Figures and Tables

**Figure 1 molecules-27-04685-f001:**
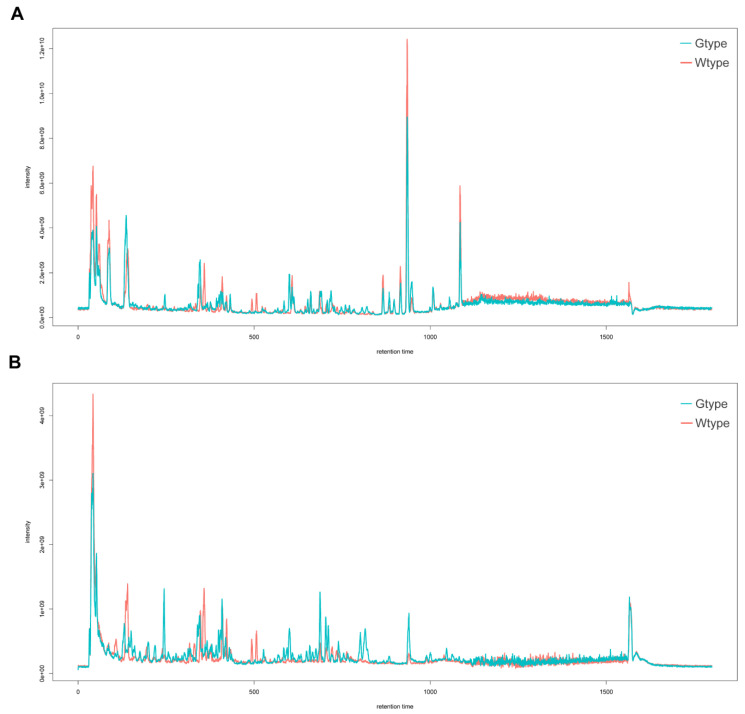
The representative total ion chromatograms of the two types of *P. sibiricum* acquired in positive (**A**) and negative (**B**) ionization modes.

**Figure 2 molecules-27-04685-f002:**
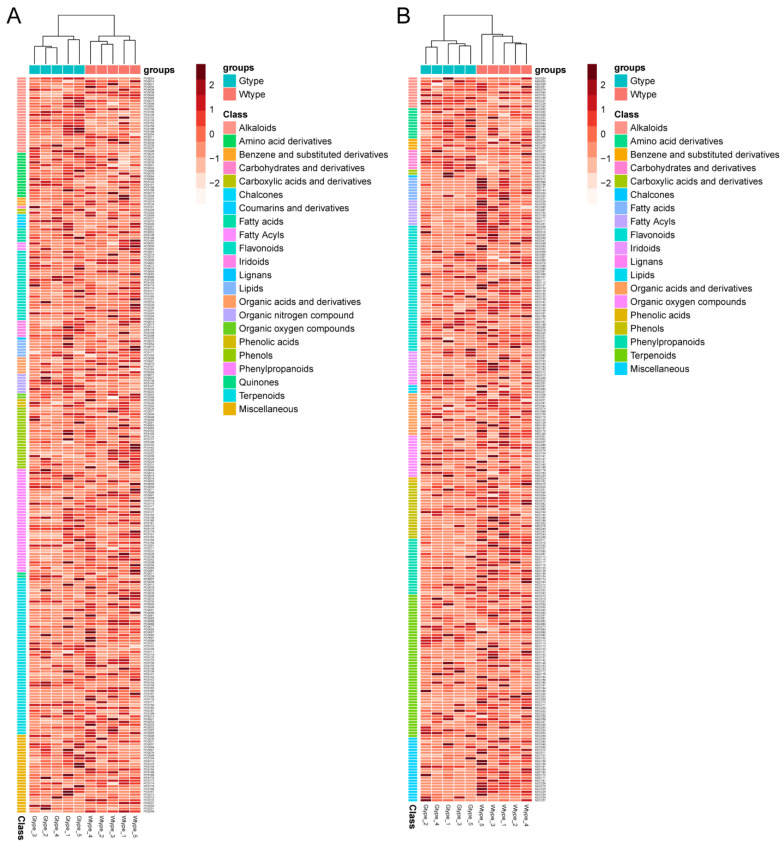
Heatmaps and hierarchical cluster analysis of the 263 tentatively identified metabolites acquired in positive ionization mode (**A**), and of the 258 tentatively identified metabolites acquired in negative ionization mode (**B**).

**Figure 3 molecules-27-04685-f003:**
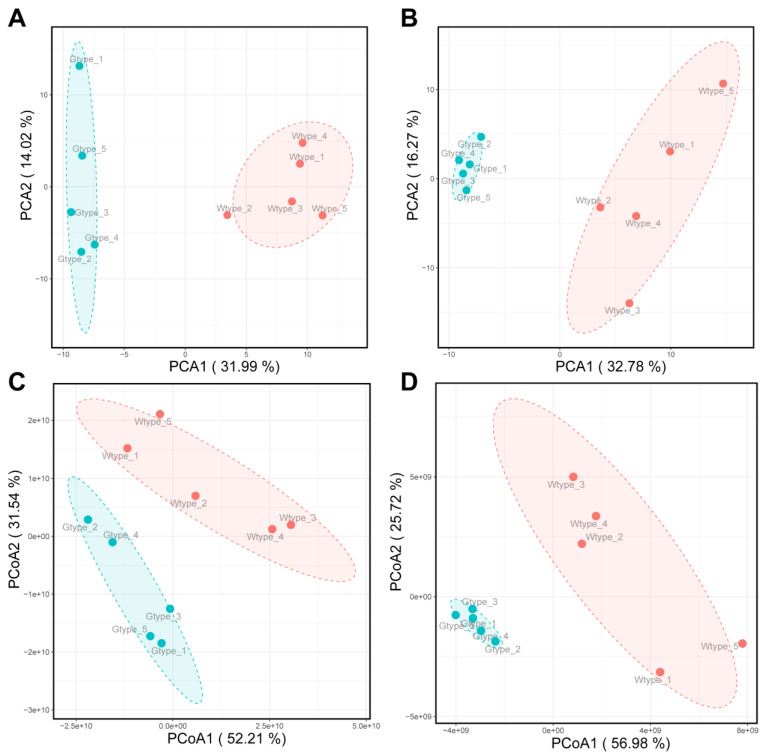
PCA score plots of the tentatively identified metabolites acquired in positive (**A**) and negative (**B**) ionization modes. PCoA score plots of the tentatively identified metabolites acquired in positive (**C**) and negative (**D**) ionization modes.

**Figure 4 molecules-27-04685-f004:**
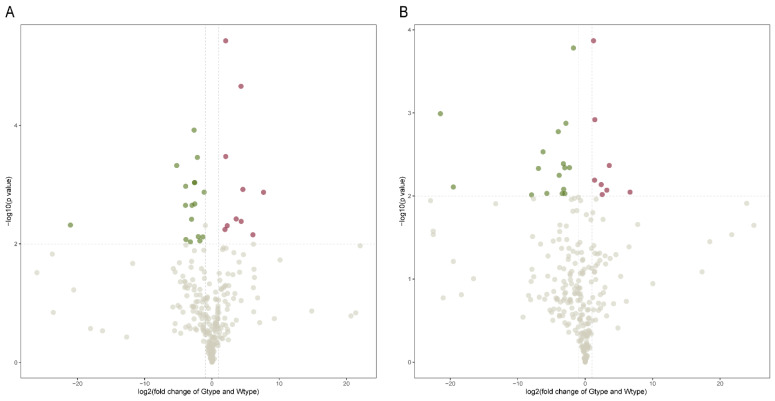
Volcano plots of significantly different metabolites of the two types of *P. sibiricum* acquired in positive (**A**) and negative (**B**) ionization modes. The negative logarithm of the *p*-value was used as the vertical axis (base 10), and the logarithm of fold change (base 2) between Gtype and Wtype samples was used as the horizontal axis. Each point in the volcano plot represents a metabolite.

**Figure 5 molecules-27-04685-f005:**
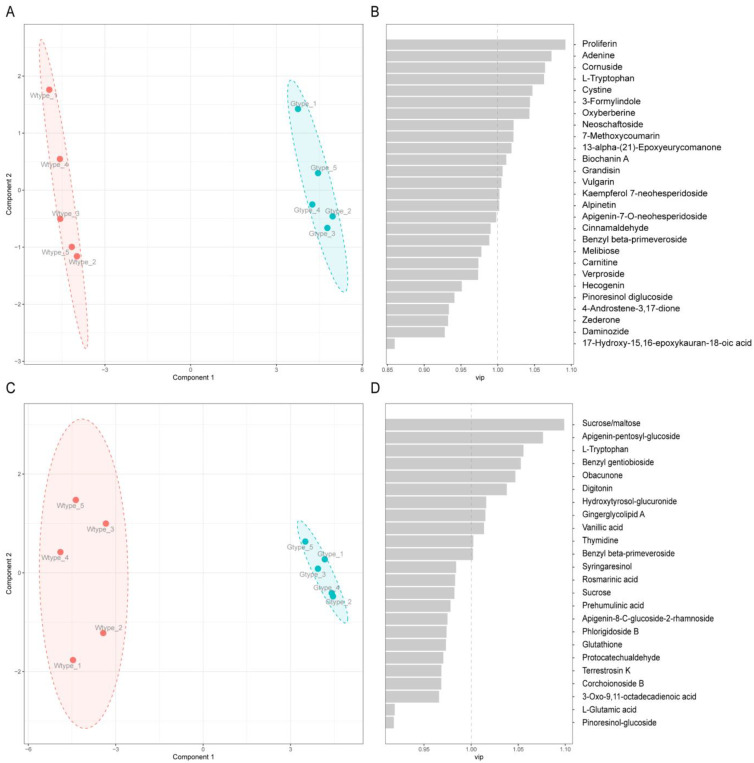
PLS-DA scatter plot (**A**) of the 27 differentially altered metabolites that were identified in positive ionization mode and their corresponding VIP values (**B**). PLS-DA scatter plot (**C**) of 24 differentially altered metabolites that were identified in negative ionization mode and their corresponding VIP values (**D**).

**Figure 6 molecules-27-04685-f006:**
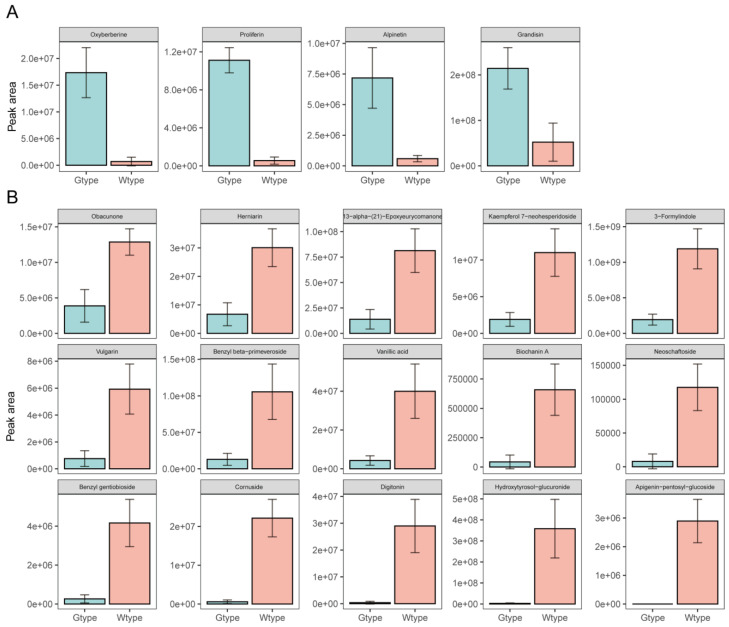
The comparison of the peak areas of the 19 differentiating metabolites belonging to phenolic compounds, terpenoids, and alkaloids, between the two types of *P. sibiricum.* (**A**) The metabolites that are significantly higher in the Gtype samples; (**B**) the metabolites that are significantly higher in the Wtype samples.

**Figure 7 molecules-27-04685-f007:**
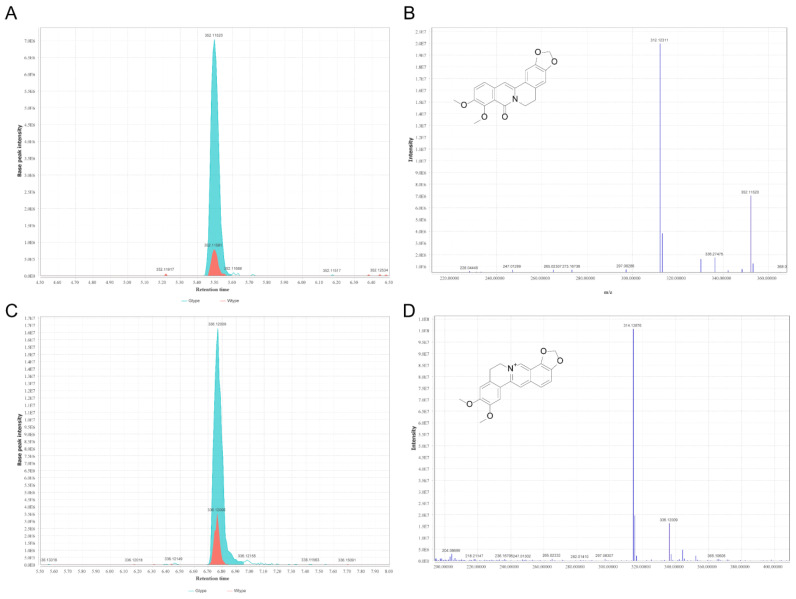
The representative extract ion chromatograms of oxyberberine (m/z 352.11555) of the two types of *P. sibiricum* acquired in positive ionization mode (**A**) and its MS2 spectra (**B**). The representative extract ion chromatograms of epiberberine (m/z 336.12009) of the two types of *P. sibiricum* acquired in positive ionization mode (**C**) and its MS2 spectra (**D**).

**Figure 8 molecules-27-04685-f008:**
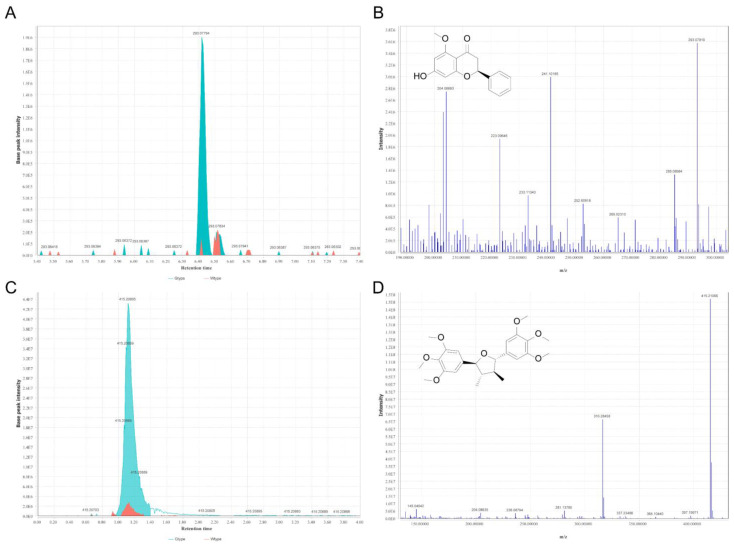
The representative extract ion chromatograms of alpinetin (m/z 293.07944) of the two types of *P. sibiricum* acquired in positive ionization mode (**A**) and its MS2 spectra (**B**). The representative extract ion chromatograms of grandisin (m/z 415.21112) of the two types of *P. sibiricum* acquired in positive ionization mode (**C**) and its MS2 spectra (**D**).

## Data Availability

Data will be provided upon request.

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
