# Peer review of "UHPLC-Q-Orbitrap-MS-Based Metabolomics Reveals Chemical Variations of Two Types of Rhizomes of Polygonatum sibiricum"

_molecules, 2022, doi:10.3390/molecules27154685_

Round 1

Reviewer 1 Report

 The manuscript developed by Luo and collaborators describes the comparative metabolic profiles of the rhizomes from two varieties of P. sibiricum (Gtype) and wide type (Wtype) by using UHPLC-Q-Orbitrap-MS based untargeted metabolomics approach. Several compounds such as alkaloids, phenols, terpenoids, amino acids, carbohydrates and lipids with pharmacological applications were identified. Using PCA, PCoA and PLS-DA modelling analysis, the authors identified some compounds as biomarkers to distinguish these varieties of P. sibiricum. In general, the manuscript is well written and structured. The data improved the knowledge about the chemical compounds of P. sibiricum. I have some considerations that are described below to improve the quality of the manuscript.

“Abstract”

Abbreviations “PCA, PCoA and PLS-DA” should be listed the first time they are mentioned.

“Material and Methods”

Have the identified compounds been deposited in any database? If yes, please cite in the manuscript.

“Results and Discussion”

In the Figure 5. Identification of biomarkers. Lactose is not a common carbohydrate found in plants.  What explains the high VIP scores calculated in the constructed PLS-DA models?

Line 283-286. "In addition to oxyberberine, alpinetin and grandisin, proliferin, a bicyclic sestertepene, was also identified as the biomarker in distinguishing the two types of P. sibiricum. However, information regarding its existence in plants and the pharmacological activity is currently lacking." I don't think the authors can identify the compounds as biomarkers but as "potential biomarkers" since additional experiments need to be done.

Figure 7 and 8. Specify in the legend which compounds refer to each ion chromatogram.

Line 212. The topic 2.4 is too long. It could be split into two sentences. Figure 7 and 8 could be in a separate topic.  

Minor

Line 130 “Several”

Line 200 “digitonin and 2”

Author Response

General comment: 

The manuscript developed by Luo and collaborators describes the comparative metabolic profiles of the rhizomes from two varieties of P. sibiricum (Gtype) and wide type (Wtype) by using UHPLC-Q-Orbitrap-MS based untargeted metabolomics approach. Several compounds such as alkaloids, phenols, terpenoids, amino acids, carbohydrates and lipids with pharmacological applications were identified. Using PCA, PCoA and PLS-DA modelling analysis, the authors identified some compounds as biomarkers to distinguish these varieties of P. sibiricum. In general, the manuscript is well written and structured. The data improved the knowledge about the chemical compounds of P. sibiricum. I have some considerations that are described below to improve the quality of the manuscript.

Response:

We thank the reviewer for the nice words about our manuscript.

Comment 1: 

“Abstract”

Abbreviations “PCA, PCoA and PLS-DA” should be listed the first time they are mentioned.

Response:

The abbreviations have been explained in the revised manuscript.

Comment 2: 

“Material and Methods”

Have the identified compounds been deposited in any database? If yes, please cite in the manuscript.

Response:

The complete list of the tentatively identified compounds can be found in Supplementary Files table S2. The raw files have not been deposited in public database.

Comment 3: 

“Results and Discussion”

In the Figure 5. Identification of biomarkers. Lactose is not a common carbohydrate found in plants.  What explains the high VIP scores calculated in the constructed PLS-DA models?

Response:

We checked our data, this compound gave an [M-H] ion of 341.10788, and MS/MS fragment ions of 89.02427, 119.03532, 179.05588, and 101.02474, which fitted the features of lactose/sucrose/maltose. Considering lactose is not commonly found in plant material, this compound is more likely to be sucrose or maltose. We have corrected this in Figure 5. Thanks for pointing this out!

Comment 4: 

Line 283-286. "In addition to oxyberberine, alpinetin and grandisin, proliferin, a bicyclic sestertepene, was also identified as the biomarker in distinguishing the two types of P. sibiricum. However, information regarding its existence in plants and the pharmacological activity is currently lacking." I don't think the authors can identify the compounds as biomarkers but as "potential biomarkers" since additional experiments need to be done.

Response:

We agree with this reviewer’s opinion. We added an additional remark for this compound as follows:

“Therefore, this compound was classified as the “potential biomarker” for distinguishing the two types of P. sibiricum.” (line 288-289)

Comment 5: 

Figure 7 and 8. Specify in the legend which compounds refer to each ion chromatogram.

Response:

The info of the compounds has been added in the legends in the revised manuscript.

Comment 6: 

Line 212. The topic 2.4 is too long. It could be split into two sentences. Figure 7 and 8 could be in a separate topic. 

Response:

We agree with this reviewer’s opinion that the session 2.4 is long. Because it covers the topics of 1) the general comparison of the differentiating phenolic compounds, terpenoids and alkaloids (Figure 6); 2) the higher abundant alkaloids in Gtype samples (Figure 7); 3) the higher abundant phenolics in Gtype samples (Figure 8); and 4) the higher abundant compounds in Wtype samples. We think it is a bit difficult to split this session into two. We would tend to keep the structure of this session as is.

Comment 7: 

Minor

Line 130 “Several”

Response:

The manuscript has been corrected accordingly.

Comment 8: 

Line 200 “digitonin and 2”

Response:

The sentences have been rephrased as follows:

“…and digitonin. Additionally, 2 metabolites were tentatively identified as alkaloids…” (line 202-203)

Reviewer 2 Report

The manuscript "UHPLC-Q-Orbitrap-MS based metabolomics chemical...." describes the study conducted by the authors on two P.sibiricum. A newly developed evergreen species (type G) and a large type (type W). A total of 263 and 258 compounds were identified. PLS.DA modeling showed that the G-type samples have a higher content of oxyberberin, proliferin, alpinetine and grandisine than the W-type sample. Although about 15 compounds are more abundant in the W-type sample based on these very interesting results the authors suggest several potential health benefits and flavor attributes of the two type of P.sibiricum rhizomes. In my opinion, the work for its content can be considered for publication on Molecules.

Author Response

General comment: 

The manuscript "UHPLC-Q-Orbitrap-MS based metabolomics chemical...." describes the study conducted by the authors on two P.sibiricum. A newly developed evergreen species (type G) and a large type (type W). A total of 263 and 258 compounds were identified. PLS.DA modeling showed that the G-type samples have a higher content of oxyberberin, proliferin, alpinetine and grandisine than the W-type sample. Although about 15 compounds are more abundant in the W-type sample based on these very interesting results the authors suggest several potential health benefits and flavor attributes of the two type of P.sibiricum rhizomes. In my opinion, the work for its content can be considered for publication on Molecules.

Response:

We thank the reviewer for the nice words about our manuscript.

Reviewer 3 Report

This study is about the comparison of the metabolic profiles of two types of P. Sibiricum via UHPLC-Q-Orbitrap-MS-based metabolomics.

Since two species have great complexity, the authors take the advantage of chemometric tools which helps find the differences and similarities. The results are better discussed with the aid of PCA, PLS-DA plots as well as dendrograms. 

I think this study will be a good example of another type of species for metabolomic profiling for the researchers studying in this field.

I find this work very encouraging and recommend its publication in its original form.

Author Response

General comment: 

This study is about the comparison of the metabolic profiles of two types of P. Sibiricum via UHPLC-Q-Orbitrap-MS-based metabolomics.

Since two species have great complexity, the authors take the advantage of chemometric tools which helps find the differences and similarities. The results are better discussed with the aid of PCA, PLS-DA plots as well as dendrograms.

I think this study will be a good example of another type of species for metabolomic profiling for the researchers studying in this field.

I find this work very encouraging and recommend its publication in its original form.

Response:

We thank the reviewer for the nice words about our manuscript.